# Effects of Glucose Levels on Inflammation and Amino Acid Utilization in Lipopolysaccharide-Induced Bovine Mammary Epithelial Cells

**DOI:** 10.3390/ani13223494

**Published:** 2023-11-13

**Authors:** Han Song, Zhiqi Lu, Kang Zhan, Osmond Datsomor, Xiaoyu Ma, Tianyu Yang, Yuhang Chen, Maocheng Jiang, Guoqi Zhao

**Affiliations:** College of Animal Science and Technology, Yangzhou University, Yangzhou 225009, China; 13273815249@163.com (H.S.); zqlu73@163.com (Z.L.); kzhan@yzu.edu.cn (K.Z.); datsomorosmond@gmail.com (O.D.); maxiaoyu0417@163.com (X.M.); 18762304846@163.com (T.Y.); chenyuhang940421@163.com (Y.C.); jmcheng1993@163.com (M.J.)

**Keywords:** dairy cow, mammary epithelial cells, glucose, amino acids, inflammation

## Abstract

**Simple Summary:**

Inflammation of bovine mammary epithelial cells (BMECs) may occur during mastitis, which affects the nutrient absorption, metabolism, and glucose and amino acid utilization of mammary epithelial cells. However, there are few studies on amino acid uptake under different nutritional conditions. Therefore, the aim of this study was to investigate the effects of lipopolysaccharide (LPS) on nutrient transport and metabolic energy supply under high- and low-glucose conditions by evaluating glucose and amino acid consumption and the expression of related genes, and to examine how different concentrations of glucose affect the proinflammatory response. BMECs were cultured in 6-well culture plates with different concentrations of glucose with or without 4 μg/mL LPS for 6 h, and then the consumption of glucose and amino acids in the supernatant was detected and the relative expression of related mRNA was determined.

**Abstract:**

Glucose and amino acids are important sources of nutrients in the synthetic milk of dairy cows, and understanding the fate of amino acids is essential to optimize the utilization of amino acids in milk protein synthesis, thereby reducing nutrient inefficiencies during lactation. The purpose of this study was to investigate the effects of LPS and different concentrations of glucose on (1) the expression of inflammatory factors and genes, (2) the glucose metabolism, and (3) amino acid utilization in BMECs. The results showed that there was an interaction (LPS × glucose, *p* < 0.05) between LPS and glucose content in the inflammatory cytokine genes (IL-6 and TNF-α) and the inflammatory regulatory genes (CXCL2, CXCL8, and CCL5). With the addition of LPS, the HG + LPS group caused downregulated (*p* < 0.05) expression of IL-6 and TNF-α, compared with the LG + LPS group. Interestingly, compared with the LG + LPS group, the HG + LPS group upregulated (*p* < 0.05) the expression of CXCL2, CXCL8, and CCL5. LPS supplementation increased (*p* = 0.056) the consumption of glucose and GLUT1 gene expression (*p* < 0.05) and tended to increase (*p* = 0.084) the LDHA gene expression of BMECs under conditions of different concentrations of glucose culture. High glucose content increased (*p* < 0.001) the consumption of glucose and enhanced (*p* < 0.05) the GLUT1, HK1, HK2, and LDHA gene expression of BMECs with or without LPS incubation, and there was an interaction (LPS × glucose, *p* < 0.05) between LPS and glucose concentrations in GLUT1 gene expression. In this study, LPS enhanced (*p* < 0.05) the consumption of amino acids such as tryptophan, leucine, isoleucine, methionine, valine, histidine, and glutamate, while high levels of glucose decreased (*p* < 0.01) consumption, except in the case of tyrosine. For histidine, leucine, isoleucine, and valine consumption, there was an interaction (LPS × glucose, *p* < 0.05) between LPS and glucose levels. Overall, these findings suggest that relatively high glucose concentrations may lessen the LPS-induced BMEC inflammatory response and reduce amino acid consumption, while low glucose concentrations may increase the demand for most amino acids through proinflammatory responses.

## 1. Introduction

Mastitis, an inflammation of the bovine mammary glands caused by changes in metabolism, environmental pathogenic microorganisms, or contagion, is one of the costliest diseases of dairy cows. Invasive pathogens are the leading cause of mastitis. *Staphylococcus aureus* and *Escherichia coli* (*E. coli*) are the two most important culprit pathogens causing mastitis in cattle [1]. Bovine mammary innate immunity relies on host cell pattern recognition receptors binding pathogen-associated molecular patterns, such as lipopolysaccharide (LPS) from *E. coli* [2,3,4]. LPS-induced bovine mammary epithelial cells (BMECs) have also been reported to increase a proinflammatory response [5,6].

Several interconnected metabolic perturbations accompany inflammation. During inflammation, the immune response increases nutrient consumption of amino acids, energy substrates, and micronutrients [7]. Our previous study demonstrated that inflammation decreased the ketogenesis-related gene expression in bovine ruminal epithelial cells [8]. In particular, glucose demand is enhanced by the immune response [9]. Kvidera et al. [10] reported that dairy cows challenged with systemic LPS exhibited extra utilization of at least 1 kg of glucose. Intramammary LPS in dairy cows increased their glucose consumption from 150 to 270 min after LPS infusion [11], which indicated that the higher glucose requirement is associated with increased hepatic glycogenolysis and gluconeogenesis, leading to an initial hyperglycemic state [12,13]. Therefore, the activation of the immune response consumes energy substrates. However, little is known about the relationship between metabolic adaptations and immune responses in dairy cows.

Glucose is a vital nutrient for all species but is especially pertinent to lactating dairy cows. Glucose generates Adenosine triphosphate via glycolysis and the Krebs cycle [14,15]. Amid the negative energy balance (NEB), glucose concentration is difficult to maintain in the inflammatory state [16]. Nevertheless, evidence indicates that amino acids play an essential role as substrates for energy metabolism and biosynthetic activities [17,18,19,20,21]. Glucose deficiency in the brain appears to increase the utilization of aminos as alternative energy substrates [22]. The mammary glands reduce milk protein synthesis, primarily of glucose and amino acids in dairy cows [23], in response to immune response [24,25]. However, few studies have investigated the potential effects of inflammation on BMECs [26,27,28]. In the present study, we hypothesized that amino acids might compensate for glucose deficiency during an activated immune response. However, the effects of glucose and inflammation on the cost of amino acids is unknown. Therefore, this study aimed to determine the effects of LPS and glucose on inflammatory response and amino acid utilization in BMECs.

## 2. Materials and Methods

All trials, plans, and methods were authorized by the Animal Ethics Committee of Yangzhou University, China. The study was conducted under the Care and Use of Laboratory Animals guidelines.

### 2.1. Experimental Design and Treatment

To explore the role of glucose in innate immunity and the content of amino acids, we prepared mammary gland tissues from three healthy Holstein dairy cows in mid lactation, which came from the Institute of Animal Culture Collection and Application, Yangzhou University. These tissues were respectively digested for 3 h by collagenase type I (Invitrogen, Shanghai, China), and cells were filtered using nylon mesh (75 μm) to obtain the BMECs. The BMECs that were isolated and cultured are described by Zhan et al. [29]. Briefly, treatments were arranged in a 2 × 2 factorial design with or without LPS (4 μg/mL) and with glucose levels of 17.5 mM (HG) or 2.5 mM (LG), and treatment groups were LG group, LG + LPS group, HG group, and HG + LPS group. In this study, LPS came from *E. coli* O55:B5 lyophilized powder (L6529, Sigma-Aldrich, St. Louis, MO, USA). The concentration of glucose was referenced in a previous study [30]. The BMECs were cultured for 12 h and then starved with Dulbecco’s Modified Eagle Medium (DMEM)/F12 without serum and glucose for 6 h. After the culture was finished, 4 μg/mL LPS was added to the culture medium containing the above two concentrations of glucose for 6 h. After culturing, the supernatant was collected to determine the content of amino acids and glucose, and the relative mRNA expression in the cells was detected using Quantitative Real-Time PCR (qRT-PCR).

### 2.2. qRT-PCR

The cells were seeded into 6-well plates at a density of 2 × 10^5^ cells/well for mRNA expression analysis. Following incubation, total RNA was extracted from incubated cells with the aid of the FastPure Cell/Tissue Total RNA Isolation Kit (RC101, Vazyme Biotech Co., Ltd., Nanjing, China) per the supplier’s directives. RNA purity and concentration were determined using an OD-1000 + Micro-Spectrophotometer, and RNA quality was verified via a 2% agarose gel electrophoresis. The OD260/280 proportion of the total RNA was revealed to be 1.9 in this research study, and the intensity of the 28S ribosomal RNA band in total RNA samples was nearly two times that of the 18S ribosomal RNA band, denoting that the total RNA was of great quality. Reverse transcription was accomplished via a reverse transcription kit (Takara, Beijing, China). In a final volume of 20 µL, the reverse transcription reaction mixtures’ constituents were 1 µg of total RNA and 1 PrimeScript reverse transcription Master Mix, and the reactions were conducted at 37 °C for 15 min. Reverse transcriptase was neutralized thermally for 5 s at 85 °C. The SYBR^®^ Premix Ex TaqTM II Kit (Takara) was employed for the qRT-PCR assays. In a final volume of 20 µL, the qRT-PCR reaction mixture comprised 1 SYBR^®^ Premix Ex TaqTM II, forward and reverse primers at 0.4 µM, and 100 ng of cDNA templates, and the reactions were carried out as follows: initial denaturation for 30 s at 95 °C, preceded by 40 cycles for 5 s at 95 °C and for 30 s at 60 °C. Before running the samples using qRT-PCR, primers were fabricated to span exon–exon junctions whenever likely, and dimer inception was assessed via the creation of melt curves subsequent to amplification to validate the existence of a lone end product. Table 1 lists the primers that were used. A negative control reaction was carried out without a cDNA sample. RefFinder (http://blooge.cn/RefFinder/ accessed on 18 March 2023), which includes Normfinder, geNorm, and the comparative ΔCT method, was used to determine the rankings of the candidate genes and the selection of the first-rank reference gene (ACTB, RPS9, and GAPDH). The ultimate ranking was determined by means of allocating an appropriate weight value to every gene and confirming the total final ranking via the geometric mean of their weight values. A lower gene geomean of ranking value was indicative of a higher expression stability. Finally, GAPDH was screened for subsequent study. The results were analyzed by using the 2^−ΔΔCt^ method, which quantifies the fold changes in mRNA levels of targeted genes [31]. All trials had three replicates.

### 2.3. Chromatographic Conditions

The standard amino acid mixture with a concentration of 2 mmol/L of each amino acid was prepared. 

Preparation of mobile phase A was as follows: weigh 8.203 g anhydrous sodium acetate, add 930 mL ultra-pure water, stir to dissolve, adjust the pH of ice acetic acid to 6.5, add 70 mL acetonitrile, mix and filter with 0.22 μm filter membrane, follow with ultrasonic degassing for 15 min. Preparation of mobile phase B was as follows: accurately measure 800 mL acetonitrile solution, add 100 mL methanol solution and 100 mL ultra-pure water, thoroughly mix, filter with 0.22 μm organic-phase filter membrane, follow with ultrasonic degassing for 15 min.

The flow rate of gradient elution was 0.7 mL/min, the column temperature was 40 °C, and the detection wavelength was 254 nm. The samples were analyzed using Waters 1525 Binary High-Performance Liquid Chromatography Pump and Waters 2998 Photodiode Array Detector with SunFire^TM^ C18 column (4.6 mm × 250 mm, 5 μm). 

Preparation of phenyl isothiocyanate acetonitrile solution was as follows: accurately measure 50 μL phenyl isothiocyanate, add 4 mL acetonitrile, shake and mix.

Preparation of triethylamine acetonitrile solution was as follows: accurately measure 1.4 mL triethylamine solution and add acetonitrile for 8.6 mL. 

Derivative of amino acid mixed standard was achieved as follows: place 200 μL mixed amino acids standard in 1.5 mL centrifuge tube, and accurately add 20 μL norleucine internal standard solution, 100 μL phenyl isothiocyanate acetonitrile solution, and 100 μL triethylamine acetonitrile solution, swirling for 15 s, and leave at room temperature for 1 h. After the derivatization, add 400 μL n-hexane, cover tightly, swirl for 30 s, stand at room temperature for 15 min, take the lower solution, filter through 0.22 μm organic-phase filter membrane, take 200 μL filtrate and add 800 μL filtered ultra-pure water, mix well, absorb 10 μL for detection.

The operation of the sample cell supernatant is the same as that of standard amino acid mixtures.

### 2.4. Statistical Analysis

Data generated were subjected to two-way ANOVA analysis, and the statistical model included the effects of LPS and glucose. Statistical significance was set at *p* < 0.05. A post hoc test was conducted using Bonferroni’s multiple comparison test (*p* < 0.05) for significant interactions. The experiments were conducted in triplicate, and all experiments were repeated at least three times. Prior to conducting the ANOVA, Levene’s test was used to ensure that the assumption of homogeneity of variances was met (*p* > 0.05).

## 3. Results

### 3.1. Effects of LPS and Glucose on Inflammation Regulation and Inflammatory Cytokine Gene Expression of BMECs

Our data showed that the interaction between the levels of LPS and glucose affected IL-6 and TNF-α gene expression, as shown in Figure 1B,C (LPS × glucose, *p* < 0.001). The LG + LPS and HG + LPS groups increased in IL-6 and TNF-α mRNA expression (*p* < 0.05), and the HG group did not show any effect in the gene expression of IL-6 and TNF-α compared to the LG group. Interestingly, HG + LPS decreased the gene expression of IL-6 and TNF-α compared to the LG + LPS group (*p* < 0.05).

For the expression of CXCL2, CXCL8, and CCL5, chemokines recruited neutrophils during the inflammation response (Figure 1D,F,G). Compared with the LG group, the LG + LPS and HG groups increased their gene expressions of CXCL2, CXCL8, and CCL5, and the highest expression of CXCL2, CXCL8, and CCL5 occurred in the HG + LPS group (LPS × glucose, *p* < 0.05). A tendency for the same response also showed up in CXCL6 (Figure 1E, LPS × glucose, *p* = 0.406), while the LG + LPS, HG, and HG + LPS groups increased their expressions of CXCL6 compared with the LG group. 

### 3.2. Effects of LPS and Glucose on Glucose Metabolism of BMECs

The expression of GLUT8 was not affected by LPS or glucose (Figure 2B, LPS × glucose, *p* = 0.536). However, the LPS-induced BMECs increased the mRNA expression of GLUT1 (Figure 2A, LPS × glucose, *p* < 0.05), which is consistent with an increase in glucose utilization. Higher glucose level without LPS led to an upregulation of GLUT1 (*p* < 0.001). The LG + LPS, HG, and HG + LPS groups increased their expressions of GLUT1 (*p* < 0.05), and the highest expression of GLUT1 occurred in the HG + LPS group, compared with the LG group. Furthermore, there was an interaction between LPS and glucose that affected GLUT1 gene expression (LPS × glucose, *p* < 0.05), which seems to support the HG group’s increased glucose utilization.

Our data showed that LPS did not affect the gene expression of HK1 (*p* = 0.332) and HK2 (*p* = 0.871), as shown in Figure 2C,D, but the incubation of glucose levels affected the mRNA expression of HK1 (*p* < 0.01) and HK2 (*p* < 0.05), and the interaction between LPS and glucose levels did not affect the gene expression of HK1 (LPS × glucose, *p* = 0.496) and HK2 (LPS × glucose, *p* = 0.513). Compared with the LG group, the HG group increased its mRNA expression of HK1 (*p* < 0.05) and HK2 (*p* < 0.05). Under HG conditions, there was the presence an LPS trend to decrease the gene expression of HK2 (*p* = 0.068). However, under LG conditions, LPS did not affect the gene expression of HK1, while it had a tendency to increase the expression of HK2 (*p* = 0.068). 

For LDHA gene expression (Figure 2E), LPS-induced BMECs had an upward tendency (*p* = 0.084), while the high concentration of glucose increased (*p* < 0.05). The LG + LPS group and the HG group had a rising trend (*p* = 0.052), while the HG + LPS group enhanced its expression (*p* < 0.05) of LDHA compared with the LG group.

The glucose uptake results are shown in Figure 2F. The concentration of glucose affected glucose consumption (*p* < 0.001), while there was a tendency to increase glucose consumption through the addition of LPS (*p* = 0.056). The amount of glucose consumed increased in the HG and HG + LPS groups (*p* < 0.05), and the highest amount of glucose consumed was in the HG + LPS group, compared with the LG group (LPS × glucose, *p* = 0.980). 

### 3.3. Effects of LPS and Glucose on Amino Acid Utilization of BMECs

To elucidate the depletion of amino acids due to inflammation and low glucose, we tested the amino acid content in the supernatant (Table 2). The interactions between LPS and the different levels of glucose affected (LPS × glucose, *p* < 0.05) histidine and branched-chain amino acids (BCAAs), such as leucin, isoleucine, and valine. Compared with the LG group, the LG + LPS group increased (*p* < 0.05) its consumption of histidine, leucin, isoleucine, and valine, while that of the HG and HG + LPS groups was reduced (*p* < 0.05). LPS incubation can take away more amino acids (*p* < 0.05), such as tryptophane, leucin, isoleucine, methionine, valine, histidine, and glutamate. Higher levels of glucose decreased (*p* < 0.01) their consumption, except for tyrosine, and the HG group consumed the fewest amino acids.

## 4. Discussion

The innate immune system represents the first line of defense in the host’s response to infection and is prepared to immediately recognize and respond to the early stages of infection [32,33]. A key component of the host’s innate immune response to infection is the upregulation of cytokine production [34,35]. IL-6 is an essential inflammatory mediator that appears to be a similar marker of early inflammation and prognosis in several diseases [36]. TNF-α is an extremely proinflammatory cytokine with injurious properties [37]. One of the most common pathogenic microorganisms causing severe acute inflammatory response leading to mastitis is *E coli* [38], which is a gram-negative bacterium. LPS is one of the most important components of the outer wall of gram-negative bacteria, which can cause inflammation in BMECs [39]. In this study, LPS-induced BMECs increased the expression of IL-1β, IL-6, and TNF-α, which is consistent with previous studies [40,41]. Zhang et al. [42] demonstrated that after BMECs were induced with LPS, the mRNA expression of TNF-α was significantly increased at all glucose concentrations, especially at low glucose concentrations, which is consistent with our results. Glucose plays a vital role in immunity; inflammation can increase catabolism and enhance glucose and amino acid demand in the bovine mammary gland [43,44,45]. Activated inflammation response enhances whole-body glucose requirements [10]. Inflammation is a biological response of the immune response. Activation of the immune response is characterized by decreased milk synthesis, maybe due to immune system nutrient consumption [23,24], especially of glucose and amino acids, in dairy cows [25]. The mammary gland is uniquely designed for the valid uptake and metabolism of nutrients, especially glucose. Furthermore, the mammary gland is the organ that consumes the most glucose [46]. Inadequate nutrient uptake and simultaneously high milk production levels may lead to an NEB in dairy cows [47,48]. The NEB takes place in the transition period until week 7 to 9 postpartum in dairy cows [49,50]. It is generally assumed that typical endocrine and metabolic adaptations occur during the NEB [48]. In addition, the immune system is impaired during periods of NEB, and the vast majority of production diseases, including mastitis, reach the greatest risk level during the transition period [51,52]. Moreover, periparturient cows are faced with more rapid bacterial growth, greater peak bacterial concentrations, and higher proinflammatory cytokine concentrations in the foremilk than mid-lactation cows [53]. However, during the NEB, glucose is maintained at lower levels, frequently accompanied by perturbation of the immune response in cattle and persistence of bacterial infections in the mammary glands, resulting in mastitis [52]. Clearance of these bacteria from the breast relies on a rapid and strong innate immune response, including the secretion of cytokines such as IL-1β and IL-6 and chemokines such as IL-8, which attract neutrophils to the site of infection [54]. In this research, LPS-induced BMECs which were cultured in high levels of glucose decreased the expression of inflammation-related genes such as IL-6 and TNF-α. This may be because higher levels of glucose are used more by immune cells to alleviate inflammation [10]. Similar research found that the addition of rumen-protected glucose could reduce the incidence of inflammation in dairy cows [55]. In the present study, higher concentrations of glucose enhanced the expression of some chemokines, such as CXCL2, CXCL8, and CCL5, while CXCL6 gene expression was also increased without a significant difference. During bovine mastitis, chemokines can recruit neutrophils to the mammary gland infected by the pathogen to relieve the inflammatory response and eliminate the pathogen [56,57]. Therefore, recruited neutrophils play a vital role in the resolution of mastitis. The present study showed that glucose was able to modulate the inflammatory response to LPS in BMECs and may further affect cellular processes or metabolism.

Glucose is vital in virtually all organisms, especially in dairy cows. Importantly, the vast majority of the glucose that is present in the blood of the lactating cow is used for lactose synthesis by the mammary gland during lactation [58] because glucose is involved in lactose synthesis, and lactose determines the amount of milk by maintaining osmotic pressure [59]. At the same time, glucose is the main fuel for cellular energy, which generates ATP via glycolysis and the Krebs cycle. During the NEB, the concentration of glucose is difficult to maintain in an inflammatory state. However, this state is quickly exhausted after the inflammatory state. Therefore, glucose plays an important role in immune response. Zhao et al. [60] reported that 20 mM of glucose increased glucose uptake in BMECs, compared to a lower glucose level. In addition, the increase in glucose consumption found in the LPS-induced group indicated that inflammation activation shifts glucose toward glycolysis or oxidation [61]. To elucidate the mechanism of glucose utilization, we detected the expression of genes related to glucose metabolism. In the bovine mammary gland, both GLUT1 and GLUT8 are vital glucose transporters [62,63]. Our data showed that the expression of GLUT1 was enhanced in the HG group or with the addition of LPS, as in the LG + LPS and HG + LPS groups, compared with the LG group. Lin et al. [64] showed that compared with no glucose, 12 mM of glucose not only increased lactose content in BMECs but also increased the expression of genes involved in glucose transport, such as GLUT1, which may be due to the fact that GLUT1 is the main glucose transporter in BMECs for lactose synthesis, and its levels are regulated by glucose. In addition, GLUT1 may be important for regulating inflammation in the skin [65]. During inflammation-induced hypoxia, the expression of GLUT1 was enhanced, which induced the upregulation of glycolysis in inflammation sites [66,67], which is consistent with the higher gene expression of LDHA. The expression of GLUT8 mRNA was not affected by different concentrations of glucose, which is consistent with a previous study [60]. It is possible that a higher level of glucose enhances GLUT1 in part by increasing inflammatory activation. Hexokinase catalyzes glucose to glucose-6-phosphate as the first step in glycolysis. Xiao et al. [68] demonstrated that hexokinase may play a vitally important role in glucose metabolism in bovine mammary glands. In this study, high levels of glucose enhanced the expression of GLUT1 and HK1, which is consistent with some previous research (12 mM) [64]. With lower levels of glucose, there was no difference in HK1 and HK2 gene expression between groups with and without LPS treatment. This may be because glucose is quickly exhausted after the inflammatory state; therefore, a change in the gene expression of HK1 and HK2 may not have been detectable under LG conditions. Interestingly, LPS-induced BEMCs increased the expression of LDHA in both the LG and HG groups, which supports the claim that inflammation activation increases glucose metabolism toward glycolysis. In the present study, BMECs had a tendency to consume more glucose when stimulated by LPS. Similar findings have been found in buffalo; serum glucose concentrations in buffalo with subclinical mastitis were significantly lower than in healthy buffalo [69]. This may also be due to the fact that inflammation increases the consumption of glucose to reduce the concentration of glucose in the serum.

Amino acids have important physiological significance and are components of proteins and substrates for the synthesis of small molecules. BCAAs are important for maintaining inflammation response [70]. Furthermore, BCAAs can be converted to α-ketoisocaproate, α-ketoisovalerate, and α-keto-β-methylvalerate, and eventually they become an intermediate in the tricarboxylic acid cycles of cholesterols such as acetyl-CoA, acetoacetate, and succinyl-CoA [71]. During inflammation activation, BCAAs can be transaminated to glutamate in order to increase glutamine synthesis [71]; this may also indirectly explain why LPS-induced BMECs consume more glutamate. Glutamine is a vital fuel for the cells of the inflammation response, contributing to inhibition of apoptosis, protein synthesis, and activation of macrophage function [70], suggesting that BCAAs may have this effects on BMECs. Studies have also shown that leucin can decrease H_2_O_2_-induced BMEC injuries by producing more ATP for energy potential [72]. The concentration of BCAAs in the plasma of cows decreases before calving and reaches its lowest level on the day of calving due to the NEB [73,74], and inflammation is often challenged at this time. In addition, amino acids are important for the body’s immune function and recovery, but their supply tends to decrease during times of stress or inflammation [75]. Therefore, we supposed that LPS-induced BMECs have a high consumption of amino acids at low glucose concentrations. The results showed that BMECs consumed more BCAAs and histidine when cultured in low concentrations of glucose with or without the induction of LPS, and the BMEC consumption of amino acids induced by LPS was the highest in low concentrations of glucose culture. The most-consumed amino acids are BCAAs and histidine, which may convert to intermediates of the tricarboxylic acid cycle to compensate for the glucose deficit. When cows encounter an NEB, ketosis may occur due to insufficient energy. Similar studies have shown that serum leucine, histidine, and valine are reduced in cows with primary ketosis compared to healthy cows, but isoleucine and glutamate concentrations are increased, which is inconsistent with our results. This may be due to the use of amino acids for the synthesis of gluconeogenesis in response to inflammation and increased glucose demand [76], which may also be a better explanation for our results. In the present study, we found that reducing glucose intake greatly increased amino acid consumption, which is consistent with our hypothesis.

## 5. Conclusions

The present study demonstrated that inflammatory response affects glucose and amino acid transport and metabolism. Our results showed that BMECs consumed more glucose for glycolysis and BCAAs induced by LPS, presumably due to the proinflammatory response. Under stimulation with LPS, the addition of glucose showed a decrease in proinflammatory cytokines and an increased recruitment of immune cells, especially neutrophils. In defiance to glucose, amino acids may provide energy compensation. However, many questions still remain. In the current study, LPS-induced BMECs with different glucose levels increased mRNA GLUT1 expression and BCAA consumption as high glucose contents decreased. In addition, the exact role of BCAAs on inflammation activation was not fully elucidated. Future research should explain the mechanisms by which GLUT1 is involved in inflammation and glucose transport and whether BCAAs play a specific role in inflammation.

## Figures and Tables

**Figure 1 animals-13-03494-f001:**
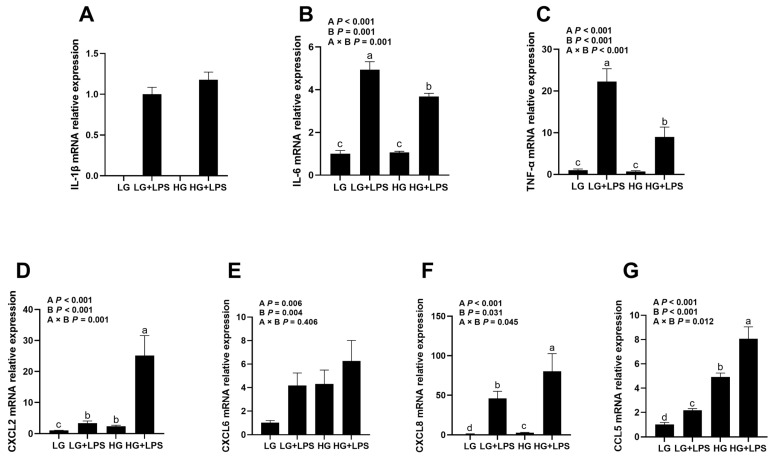
Effects of glucose concentrations on inflammation regulation and inflammatory cytokine gene expression of LPS-induced BMECs. BMECs were treated with 4 μg/mL LPS and different concentrations of glucose (2.5 and 17.5 mM). After incubation, the cells were collected for qRT-PCR. (**A**) IL-1β, (**B**) IL-6, (**C**) TNF-α, (**D**) CXCL2, (**E**) CXCL6, (**F**) CXCL8, and (**G**) CCL5. LG, low-glucose (2.5 mM) DMEM; LG + LPS, low-glucose (2.5 mM) DMEM with 4 μg/mL LPS; HG, high-glucose (17.5 mM) DMEM; HG + LPS, high-glucose (17.5 mM) DMEM with 4 μg/mL LPS; A, LPS addition; B, glucose content; A × B, interaction between LPS and glucose content. Different lowercase letters (a–d) in the bar chart indicate significant differences (*p* < 0.05).

**Figure 2 animals-13-03494-f002:**
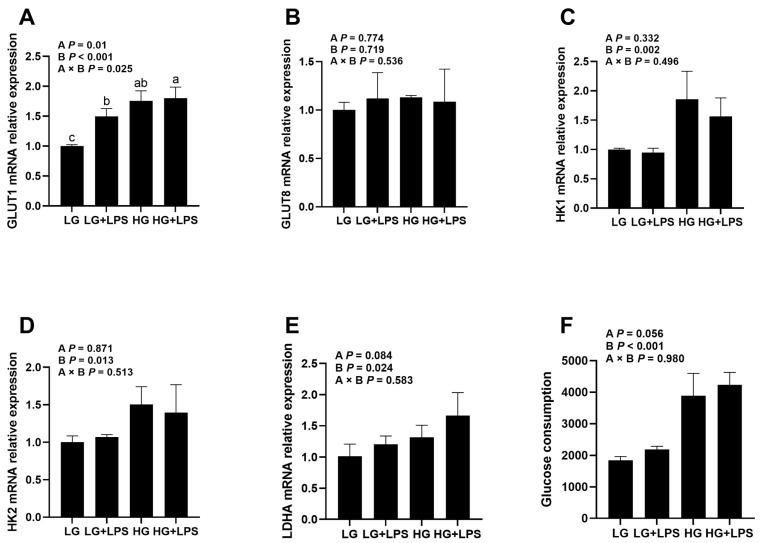
Effects of glucose concentrations on glucose metabolism gene expression and glucose consumption of LPS-induced BMECs. BMECs were treated with 4 μg/mL LPS and different concentrations of glucose (2.5 and 17.5 mM). After incubation, the cells were collected for qRT-PCR and the supernatant was collected to determine the content of glucose. (**A**) GLUT1, (**B**) GLUT8, (**C**) HK1, (**D**) HK2, (**E**) LDHA, and (**F**) glucose consumption. LG, low-glucose (2.5 mM) DMEM; LG + LPS, low-glucose (2.5 mM) DMEM with 4 μg/mL LPS; HG, high-glucose (17.5 mM) DMEM; HG + LPS, high-glucose (17.5 mM) DMEM with 4 μg/mL LPS; A, LPS addition; B, glucose content; A × B, interaction between LPS and glucose content. Different lowercase letters (a–c) in the bar chart indicate significant differences (*p* < 0.05).

**Table 1 animals-13-03494-t001:** Primers for real-time PCR analyses.

Gene	Primer Sequence, 5′ to 3′	Accession Number	Size (bp)
GAPDH	F: GGGTCATCATCTCTGCACCT R: GGTCATAAGTCCCTCCACGA	NM_001034034.2	176
IL-1β	F: CAGTGCCTACGCACATGTCT R: AGAGGAGGTGGAGAGCCTTC	NM_174093.1	209
IL-6	F: CACCCCAGGCAGACTACTTC	NM_173923.2	129
	R: TCCTTGCTGCTTTCACACTC		
TNF-α	F: GCCCTCTGGTTCAGACACTC R: AGATGAGGTAAAGCCCGTCA	NM_173966.3	192
CXCL2	F: CCCGTGGTCAACGAACTGCGCTGC R: CTAGTTTAGCATCTTATCGATGATT	NM_174299.3	204
CXCL6	F: TGAGACTGCTATCCAGCCG R: AGATCACTGACCGTTTTGGG	NM_174300.2	193
CXCL8	F: TGGGCCACACTGTGAAAAT R: TCATGGATCTTGCTTCTCAGC	NM_173925.2	136
CCL5	F: CTGCCTTCGCTGTCCTCCTGATG R: TTCTCTGGGTTGGCGCACACCTG	NM_175827	217
GLUT1	F: CGTGCTCCTGGTTCTGTTCT R: GGAACAGCTCCTCAGGTGTC	NM_001274304.1	137
GLUT8	F: AGTGACTGCCCGTCCTTGCT R: TGCTGTCCTGGCTCCTGACT	NM_201528.1	133
HK1	F: GTGTGCTGTTGATAATCTCC R: AATAACTGTTGGACGAATGC	NM_001012668.2	149
HK2	F: AAGATGCTGCCCACCTACG R: TCGCTTCCCGTTCCGCACA	XM_015473383.2	123

F, forward; R, reverse.

**Table 2 animals-13-03494-t002:** Effects of glucose concentrations on amino acid uptake of LPS-induced BMECs.

	Treatment ^1^					*p*-Value ^2^		
Item	LG	LG + LPS	HG	HG + LPS	SEM	A	B	A × B
Lysine	0.2938	0.3566	0.0989	0.1184	0.034	0.077	<0.001	0.318
Tryptophan	0.0312	0.0366	0.0116	0.0153	0.003	0.025	<0.001	0.633
Phenylalanine	0.1141	0.1606	0.0599	0.0655	0.014	0.192	0.004	0.295
Leucine	0.1683 ^b^	0.2226 ^a^	0.0999 ^d^	0.1312 ^c^	0.015	<0.001	<0.001	0.034
Isoleucine	0.2620 ^b^	0.3409 ^a^	0.1041 ^c^	0.1162 ^c^	0.030	0.004	<0.001	0.021
Methionine	0.0240	0.0276	0.0297	0.0397	0.001	0.004	0.001	0.101
Valine	0.2001 ^b^	0.2785 ^a^	0.1081 ^c^	0.1226 ^c^	0.023	0.004	<0.001	0.024
Tyrosine	0.0589	0.0626	0.0594	0.0657	0.002	0.453	0.782	0.842
Proline	0.1033	0.1207	0.0483	0.0516	0.009	0.136	<0.001	0.292
Threonine	0.2347	0.2685	0.1427	0.1522	0.017	0.219	<0.001	0.475
Arginine	0.3285	0.3367	0.1175	0.1495	0.032	0.445	<0.001	0.647
Histidine	0.0960 ^b^	0.1786 ^a^	0.0423 ^c^	0.0467 ^c^	0.016	0.001	<0.001	0.002
Glycine	0.1761	0.2067	0.0602	0.0674	0.020	0.103	<0.001	0.288
Serine	0.1319	0.1427	0.0579	0.0714	0.011	0.266	<0.001	0.857
Glutamate	0.0173	0.0286	0.0120	0.0190	0.001	0.001	0.003	0.263

^1^ LG = low-glucose (2.5 mM) DMEM; LG + LPS = low-glucose (2.5 mM) DMEM with 4 μg/mL LPS; HG = high-glucose (17.5 mM) DMEM; HG + LPS = high-glucose (17.5 mM) DMEM with 4 μg/mL LPS. ^2^ A = LPS addition; B = glucose content; A × B = interaction between LPS and glucose content. Different lowercase letters (a–d) in the bar chart indicate significant differences (*p* < 0.05).

## Data Availability

Data are contained within the article.

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
