# Peer review of "Effects of Glucose Levels on Inflammation and Amino Acid Utilization in Lipopolysaccharide-Induced Bovine Mammary Epithelial Cells"

_animals, 2023, doi:10.3390/ani13223494_

Round 1
Reviewer 1 Report
Comments and Suggestions for Authors
The article is original and very interesting.
The topic is very relevant, the aim of the study being to investigate the effects of LPS on nutrient transport and metabolic energy supply under high and low glucose conditions by evaluating glucose and amino acid consumption and expression of related genes, and how different concentrations of glucose affect the pro-inflammatory response. The study demonstrated that inflammatory response affects glucose and amino acids transport and metabolism, BMECs consumed more glucose to glycolysis and BCAAs induced by LPS, presumably due to the pro-inflammatory response. The addition of glucose showed a decrease in proinflammatory cytokines and increased recruitment of immune cells, especially neutrophils.
The conclusions are consistent with the evidence and arguments presented.
The references are appropriate, including some very relevant author’s previous experience in the field.
1. I suggest small spelling/editing corrections, including carefull following of Instructions of authors, especially in References Section.
2.Results from original Tables should be illustrated with some original figures from Rt-PCR and cell cultures
Author Response
- I suggest small spelling/editing corrections, including carefull following of Instructions of authors, especially in References Section.
R: Thank you very much for your suggestions. We made some spelling corrections and followed the author's instructions to the letter, and all the reference formats are checked and modified.
- Results from original Tables should be illustrated with some original figures from Rt-PCR and cell cultures.
R: Thank you very much for your suggestions. We have presented the results of Rt-PCR and cell cultures in the form of pictures.

Reviewer 2 Report
Comments and Suggestions for Authors
The objective of the authors was to determine a potential effect of lipopolysaccharide and glucose on inflammatory response and amino acid utilization in mammary cells of cows.
Major issues
First point: please do not use some many abbreviations, it becomes irritating for the reader! Abbreviations must be kept to a minimum number during the revision. Maintaining abbreviations will lead to a recommendation of outright rejection during the next evaluation.
2.1. Please describe the criteria for selection of the three cows.
Please describe the procedure for obtaining the epithelial cells.
Table 1 does not include all the details of the PCRs. These must be added.
2.4. Why using ANOVA? Did you prove that data were normally distributed? Please consider using non-parametric techniques if needed.
Results. Lack of visualization makes the manuscript weak. Please add graphs to summarize the findings.
Discussion. I) The discussion could be better divided in two sub-sections to allow better flow of ideas. II) The discussion does not go into explaining the full array of the findings and in substantial depth, please expand. III) Please add a paragraph to compare with relevant findings in other ruminants (e.g., buffaloes) and also please add the appropriate references.
Overall. Significant revision is necessary to improved the manuscript and subsequently new assessment must be performed.
Author Response
Major issues
- First point: please do not use some many abbreviations, it becomes irritating for the reader! Abbreviations must be kept to a minimum number during the revision. Maintaining abbreviations will lead to a recommendation of outright rejection during the next evaluation.
R: Thank you very much for your suggestions. We are sorry that there are too many abbreviations in the manuscript and I have tried to minimize the number of abbreviations. Finally, we have only retained the abbreviations of LPS, BMECs, NEB, LG, LG + LPS, HG, HG + LPS, DMEM, qRT-PCR, and BCAAs. We are sorry for that we are not sure if it's because too many group names like LG, LG + LPS, HG, and HG + LPS use abbreviations. If it is due to too many abbreviations in the group name, we can modify it.
- Please describe the criteria for selection of the three cows.
R: Thank you very much for your suggestions. The three healthy Chinese Holstein dairy cows in mid-lactation with similar body weight and lactation performance came from the experimental farm of Yangzhou University.
- Please describe the procedure for obtaining the epithelial cells.
R: Thank you very much for your suggestions. We got mammary tissue from these cows and these tissues were digested by collagenase type I (Invitrogen, Shanghai, China) respectively for 3 h, and cells were filtered by nylon mesh (75 μm) to obtain the BMECs. BMECs were seeded in DMEM/F12 medium supplemented with 10% fetal bovine serum, 4 mm/L glutamine, 1× insulin, transferrin, sodium selenite (10 μg/mL insulin, 5.5 μg/mL transferrin, 0.0067 μg/mL sodium selenite, Invitrogen, Shanghai, China), 15 ng/mL epidermal growth factor (Peprotech,Shanghai, China), 1 μg/mL hydrocortisone, and 4 μg/mL prolactin (Sigma-Aldrich, Shanghai, China). We describe how we got the BMECs in lines 90-92 of the manuscript. The specific method is in the previous study [1-2].
- Table 1 does not include all the details of the PCRs. These must be added.
R: Thank you very much for your suggestions. We are sorry for not including all PCR details in Table 1. We have searched and supplemented the size of all primers in Table 1.
- Why using ANOVA? Did you prove that data were normally distributed? Please consider using non-parametric techniques if needed.
R: Thank you very much for your suggestions. We have consulted with a statistician at Yangzhou University. Therefore, we checked the data and made a Levene's Test of Equality of Error Variances of all the data in this experiment, and the results showed that the P-values of all test indicators were greater than 0.05, whether based on the mean or the median, this means that ANOVA can be performed. Thus, we used two-way ANOVA analysis to analyze the results. In addition, we have supplemented the content in lines 168-169 of the manuscript.
- Results. Lack of visualization makes the manuscript weak. Please add graphs to summarize the findings.
R: Thank you very much for your suggestions. We are sorry that the results in the manuscript are not shown in graphic form. We have presented some of the results in the form of drawings to enrich the manuscript, so that the results can be better presented, and we add a graph in line 47 to summarize the results to make the paper more visual.
- The discussion could be better divided in two sub-sections to allow better flow of ideas.
R: Thank you very much for your suggestions. After our consideration, we divided the discussion into three sections to make the paper more logical. The inflammatory response, glucose metabolism, and amino acid consumption in different treatments were discussed respectively.
- The discussion does not go into explaining the full array of the findings and in substantial depth, please expand.
R: Thank you very much for your suggestions. We have a better discussion and interpretation of the results. In addition, we expanded and supplemented the discussion in line 255-257, 281-287, 328-335, 345-349, 351-354, and 356-361.
- Please add a paragraph to compare with relevant findings in other ruminants (e.g., buffaloes) and also please add the appropriate references.
R: Thank you very much for your suggestions. We wrote about some discussion on buffaloes. In addition, we discussed the similar research about buffaloes in line 329-333.
Abbreviation
LPS: lipopolysaccharide; BMECs: bovine mammary epithelial cells; NEB: negative energy balance; LG: low-glucose (2.5 mM) DMEM; LG + LPS: low-glucose (2.5 mM) DMEM with 4 μg/mL LPS; HG: high-glucose (17.5 mM) DMEM; HG + LPS: high-glucose (17.5 mM) DMEM with 4 μg/mL LPS; DMEM: Dulbecco’s modified Eagle medium; qRT-PCR: Quantitative Real Time-PCR; BCAAs: branched-chain amino acids.
Reference
[1] Zhan, K.; Yang, T.; Feng, B.; Zhu, X.; Chen, Y.; Huo, Y.; Zhao, G. The protective roles of tea tree oil extracts in bovine mammary epithelial cells and polymorphonuclear leukocytes. J. Anim. Sci. Biotechnol. 2020, 11, 62, doi:10.1186/s40104-020-00468-9.
[2] Gong, X.X.; Su, X.S.; Zhan, K.; Zhao, G.Q. The protective effect of chlorogenic acid on bovine mammary epithelial cells and neutrophil function. J. Dairy Sci. 2018, 101, 10089–10097, doi:10.3168/jds.2017-14328.

Round 2
Reviewer 1 Report
Comments and Suggestions for Authors
Since the authors have made all the corrections suggested, I recommend the acceptance of the article
Reviewer 2 Report
Comments and Suggestions for Authors
The authors have corrected the various flaws indicated in the previous version and hence have improved the revised manuscript.
I have no further comments.